# Factors to Consider to Study Preductal Oxygen Saturation Targets in Neonatal Pulmonary Hypertension

**DOI:** 10.3390/children9030396

**Published:** 2022-03-11

**Authors:** Heather Siefkes, Sherzana Sunderji, Jessica Vaughn, Deepika Sankaran, Payam Vali, Pranjali Vadlaputi, Sage Timberline, Avni Bhatt, Daniel Tancredi, Satyan Lakshminrusimha

**Affiliations:** 1Department of Pediatrics, University of California, Davis, Sacramento, CA 95817, USA; ssunderji@ucdavis.edu (S.S.); jjvaughn@ucdavis.edu (J.V.); dsankaran@ucdavis.edu (D.S.); pvali@ucdavis.edu (P.V.); ppvadlaputi@ucdavis.edu (P.V.); stimberline@ucdavis.edu (S.T.); djtancredi@ucdavis.edu (D.T.); 2School of Medicine, University of California, Davis, Sacramento, CA 95817, USA; avbhatt@ucdavis.edu

**Keywords:** persistent pulmonary hypertension of the newborn (PPHN), oxygen saturation, hypoxic respiratory failure, pulmonary vascular resistance, randomized trial, study protocol

## Abstract

There are potential benefits and risks to the infant with higher and lower oxygen saturation (SpO_2_) targets, and the ideal range for infants with pulmonary hypertension (PH) remains unknown. Targeting high SpO_2_ can promote pulmonary vasodilation but cause oxygen toxicity. Targeting lower SpO_2_ may increase pulmonary vascular resistance, especially in the presence of acidosis and hypothermia. We will conduct a randomized pilot trial to compare two ranges of target preductal SpO_2_ in late-preterm and term infants with hypoxic respiratory failure (HRF) and acute pulmonary hypertension (aPH) of the newborn. We will assess the reliability of a newly created HRF/PH score that could be used in larger trials. We will assess trial feasibility and obtain preliminary estimates of outcomes. Our primary hypothesis is that in neonates with PH and HRF, targeting preductal SpO_2_ of 95–99% (intervention) will result in lower pulmonary vascular resistance and pulmonary arterial pressures, and lower the need for pulmonary vasodilators (inhaled nitric oxide—iNO, milrinone and sildenafil) compared to targeting SpO_2_ at 91–95% (standard). We also speculate that a higher SpO_2_ target can potentially induce oxidative stress and decrease response to iNO (oxygenation and pulmonary vasodilation) for those patients that still require iNO in this range. We present considerations in planning this trial as well as some of the details of the protocol design (Clinicaltrials.gov (NCT04938167)).

## 1. Introduction

Successful transition at birth is dependent on establishment of lungs as the organ of gas exchange [1]. Breathing at birth and an increase in alveolar oxygen tension (PAO_2_) leads to an 8–10 fold increase in pulmonary blood flow with a marked reduction in pulmonary vascular resistance (PVR) [2]. Failure to decrease PVR at birth results in hypoxemic respiratory failure (HRF) and acute pulmonary hypertension (aPH) of the newborn [3]. Hypoxic pulmonary vasoconstriction exacerbates PH by increasing PVR [4]. However, administration of excess oxygen can lead to free radical formation, increase pulmonary vascular reactivity and reduce response to pulmonary vasodilators such as inhaled nitric oxide (iNO) [5,6,7]. There is potential for poor outcomes at both higher and lower oxygen saturations (SpO_2_) measured by pulse-oximetry in aPH, with the ideal target SpO_2_ range being unknown. Guidelines from the American Heart Association (AHA) and American Thoracic Society (ATS) suggest that extreme hyperoxia (fraction of inspired oxygen (F_i_O_2_) > 0.6) may aggravate lung injury, and be ineffective owing to extrapulmonary shunting [8]. The European Pediatric Pulmonary Vascular Disease Network (EPPVDN) recommends maintaining preductal SpO_2_ between 91 and 95% when persistent PH of the newborn (PPHN) is suspected or established [9]. However, there are no clinical studies evaluating optimal SpO_2_ targets in newborns with PH. Recent animal studies suggest that targeting higher SpO_2_ (95–99%) may result in lower PVR in the presence of acidosis [10]. We are conducting a pilot randomized trial of two SpO_2_ targets in neonates with HRF and PH. Here we provide the background supporting this trial, study objectives and the protocol.

Preterm infants and target SpO_2_: Large multicenter randomized trials in preterm infants (including approximately 5000 extremely preterm infants < 28 weeks gestational age (GA) at birth) comparing a target SpO_2_ of 85–89% to 91–95% have been conducted [11,12]. These trials did not show a statistically significant difference in primary combined outcome of death or neurodevelopmental impairment at ~2 years (53.5 vs. 51.6%, respectively, *p* = 0.21) but showed an increase in mortality in the 85–89% (lower) target SpO_2_ arm (19.9 vs. 17.1%, respectively, *p* = 0.01) [13]. Treatment for retinopathy of prematurity (ROP) was administered to 10.9% of infants in the lower SpO_2_ target group and 14.9% in the higher target group (*p* < 0.001). A previous randomized trial in 358 preterm infants < 30 weeks gestation at birth and on supplemental oxygen at 32 weeks postmenstrual age, comparing 91–94% vs. 95–98% target SpO_2_, did not show a statistically significant difference in mortality after randomization (3 vs. 5%, respectively, *p* = 0.41). However, there was a trend towards higher death due to pulmonary causes (0.6 vs. 3.3%, *p* = 0.12) with 95–98% compared to 91–94% SpO_2_ [14].

Term infants with PH and target SpO_2_: There are no studies evaluating optimal SpO_2_ targets in term infants with HRF or PH. At term gestation, ROP is no longer a clinical concern. EPPVDN recommends maintaining SpO_2_ between 91 and 95% when PH is suspected or established in a newborn [9]. The Canadian guidelines for management of PH in infants with CDH recommend a preductal SpO_2_ target range of 85–95% [15]. These recommendations are based on expert opinions and not on clinical evidence. A survey of 492 neonatologists evaluated SpO_2_ target preferences during management of aPH in term infants [16]. Seventy percent of respondents selected an SpO_2_ target of ≥95% which is contrary to published guidelines/expert opinion and emphasizes the need for a clinical study [16]. Interestingly, physicians with extracorporeal life support (ECLS) experience aimed towards lower SpO_2_ targets, perhaps to lower the need for ECLS based on SpO_2_ goal alone.

Animal studies evaluating SpO_2_ in PH/HRF: Animal studies suggest benefits in targeting SpO_2_ in the mid-90s. Rudolph and Yuan studied newborn calves and demonstrated that a partial arterial oxygen tension (PaO_2_) < 45 mmHg was associated with a rapid increase in PVR [4]. Acidosis exacerbated hypoxic vasoconstriction [4]. Two recent studies in lamb models have evaluated optimal preductal SpO_2_ targets in PH/HRF. In a PH model induced by antenatal ductal ligation (without any parenchymal lung disease), preductal SpO_2_ in the mid-90s was associated with low PVR [5]. In lambs with meconium aspiration syndrome (MAS), acidosis and HRF, preductal SpO_2_ with a target of 95–99% (but with actual achieved SpO_2_ interquartile range of 93–97%) achieved lower PVR compared to a target of 90–94% (Figure 1). Some lambs that received 100% inspired oxygen and achieved 100% (median, IQR 98–100%) SpO_2_ had a trend towards higher PVR and lower pulmonary blood flow (Qp) compared to lambs with target SpO_2_ of 95–99% (with lower exposure to F_i_O_2_), possibly due to vascular free radical injury [10]. These data suggest that there are risks to both hypoxemia and hyperoxemia, but that normoxemia or the ideal SpO_2_ range may be higher than our current standard practice, especially in the presence of acidosis (a common accompaniment to hypoxia in HRF and aPH).

### 1.1. Rationale for Higher (95–99%) SpO_2_ in PH

a.Acidosis: Infants with aPH often have associated respiratory and metabolic acidosis that can exacerbate hypoxic pulmonary vasoconstriction [4]. Maintaining a higher SpO_2_ target (95–99%) may limit hypoxic pulmonary vasoconstriction when pH is <7.2.b.Optimal F_i_O_2_: We have previously shown in the lamb model of asphyxia, MAS and PH, that targeting 95–99% SpO_2_ (achieving a median of 97%) resulted in lower PVR compared to the 90–94% target (achieving a median of 92%—Figure 1). This reduction in PVR was not associated with a statistically significant difference in preductal PaO_2_ (56 ± 11 mmHg with 90–94% target and 58 ± 19 mmHg with 95–99% target SpO_2_—Figure 1). However, the mean F_i_O_2_ to achieve 95–99% SpO_2_ was significantly higher than the 90–94% group (0.5 ± 0.21 vs. 0.29 ± 0.17) [10]. Given the importance of alveolar PAO_2_ in mitigating hypoxic pulmonary vasoconstriction [17], we speculate that F_i_O_2_ (in addition to SpO_2_ or PaO_2_) plays an important role in pulmonary vasodilation in PH.c.Skin pigmentation, race and SpO_2_: Skin pigmentation can underestimate hypoxemia by pulse oximetry [18,19]. In neonates the discrepancy between arterial saturation (SaO_2_) and pulse oximetry (SpO_2_) is higher in Black infants compared to white infants, especially when SpO_2_ is below 95%. The incidence of occult hypoxemia (defined as SaO_2_ < 85% when SpO_2_ is ≥90%) is more common in Black infants (9.2% of samples) compared to white infants (7.7%) [19]. Targeting SpO_2_ in the low 90s might increase the risk of occult hypoxemia in infants with darker skin. Interestingly, conditions exacerbated by hypoxemia (such as necrotizing enterocolitis—NEC) and aPH are more common in Black infants, and conditions exacerbated by hyperoxia (such as ROP) are more common among white infants (Figure 2) [20,21,22,23].d.Therapeutic hypothermia: Infants with moderate to severe hypoxic ischemic encephalopathy (HIE) undergoing therapeutic hypothermia exhibit features of aPH (~25% of neonates cooled down to a core temperature of 33.5 °C and ~34% of neonates cooled to 32 °C received iNO) [24]. Konduri et al. conducted a trial on iNO use in aPH prior to the advent of therapeutic hypothermia for HIE. Among all term neonates with aPH, 34/299 (11.4%) needed ECMO. Low Apgar scores (<3 at 1 min) were present in 26.1% of these patients with aPH, and perinatal aspiration syndrome (44%) was the leading cause of aPH [25]. In a study by Shankaran et al., 22/105 infants with HIE on iNO for aPH needed ECLS (21%), suggesting that the response to iNO might be impaired in HIE and hypothermia. HIE is often associated with left ventricular dysfunction, which can lead to pulmonary vascular congestion from increased left atrial and pulmonary venous pressure resulting in a poor response to iNO [26,27,28]. In addition, during whole body hypothermia, the hemoglobin–oxygen dissociation curve shifts to the left (Figure 3), resulting in higher SpO_2_ values for the same PaO_2_ range. Targeting 91–95% preductal SpO_2_ might lead to hypoxemia (lower PaO_2_), increased PVR and increased need for ECLS. Finally, the use of IV vasodilators such as milrinone may be associated with severe systemic hypotension during hypothermia, and may contribute to an increased need for ECLS [29]. The optimal target SpO_2_, physiologic basis of hemodynamic and oxygenation response to iNO, sildenafil and milrinone, during whole-body hypothermia are not known.

### 1.2. Rationale for Standard (91–95%) SpO_2_ Range in PH

a.Response to pulmonary vasodilators and target SpO_2_: The target SpO_2_ that results in optimal vasodilation in response to iNO, sildenafil or milrinone is not known. Nitric oxide reacts with superoxide anions to form toxic peroxynitrite (Figure 4). The bioavailability of iNO is determined by local concentration of superoxide anions [30]. Ventilation with 100% oxygen increases superoxide anion production in pulmonary arterial smooth muscle cells and impairs response to iNO in lambs [5,31]. Gitto et al. ventilated 60 term neonates with aPH with an initial F_i_O_2_ of 0.45 or 0.8. All infants received iNO. Serum IL-6, IL-8 and TNF-α levels were measured over 72 h. Infants in the 0.45 FiO_2_ group showed progressive decrease in these inflammatory markers. Infants in the 0.8 F_i_O_2_ group saw an increase in serum IL-6, IL-8 and TNF-α levels [32]. The reduction in oxygenation index (mean airway pressure × F_i_O_2_ × 100 ÷ PaO_2_) was similar between both groups. These findings suggest a combination of high F_i_O_2_ and iNO can trigger inflammatory cytokines but does not alter improvement in oxygenation [32].b.Oxygen toxicity: Use of 100% inspired oxygen is associated with free radical formation in animal models and is associated with reduced response to pulmonary vasodilators such as iNO. The combination of iNO with high F_i_O_2_ is proinflammatory due to oxidant injury. However, iNO can exhibit anti-inflammatory and antioxidant effects when F_i_O_2_ is low. There are no studies evaluating different target SpO_2_ ranges with response to iNO.

There is no evidence-based optimal SpO_2_ target in aPH. While 100% inspired oxygen and SpO_2_ of 100% with PaO_2_ values > 150 mmHg have been associated with oxidative stress and impaired response to iNO in animal studies, there are no studies showing different oxidative stress levels or any other negative effects with targeting SpO_2_ in the 91–99% range. To our knowledge, this will be the first clinical trial evaluating two SpO_2_ target ranges (91–95% and 95–99%) in late-preterm and term infants with PH.

## 2. Materials and Methods

We will conduct a randomized nonblinded pilot trial to compare two ranges of target preductal SpO_2_ in late-preterm and term infants with HRF and PH. We will assess the reliability of an HRF/PH score (Figure 5) that could be used in larger clinical trials. We will also assess trial feasibility and obtain preliminary estimates of outcomes. Our primary hypothesis is that in neonates with PH and HRF, targeting preductal SpO_2_ of 95–99% (intervention) will result in lower PVR and lower need for non-oxygen-based pulmonary vasodilators (iNO, milrinone and sildenafil) compared to a target of 91–95% (standard). We also hypothesize that although the need for iNO and other vasodilators will be lower in the 95–99% SpO_2_ target range, when these vasodilators are utilized, the pulmonary vasodilator and oxygenation response will be lower in the intervention group (95–99% target) compared to standard group (91–95% target).

### 2.1. Objectives/Specific Aims

Aim 1: Understand the variation of the HRF/PH score among term and late-preterm neonates and inter-rater reliability of the score for future trial planning. We hypothesize that the HRF/PH score will have a substantial between-infant variance component relative to the within-infant and between-rater variance components, supporting its inter-rater reliability and informing the design of future trials.

Aim 2: Estimate the relationship of the HRF/PH score to outcomes such as duration of mechanical ventilation, pulmonary vasodilator use, incidence of ECLS, and survival.

Aim 3: Assess the feasibility of future trials in regard to patient enrollment, withdrawal or crossover between standard and intervention arms.

Aim 4: Obtain preliminary estimates of the intervention’s impact on the HRF/PH score, length of stay, duration of mechanical ventilation and need for ECLS. To assist with future trial design, we will estimate the impact of the intervention on outcomes. Even if the HRF/PH score is determined to be invalid or have poor reliability, we will be able to measure the impact of SpO_2_ target on PH, and above-mentioned outcomes to assist with future larger, definitive multicenter trial design.

### 2.2. Study Design

This is a randomized single-center pilot trial. Neonates with evidence of HRF and PH will be randomized to one of two SpO_2_ target ranges, 95–99% (intervention) vs. 91–95% (standard). This pilot trial will permit development and validation of a novel HRF/PH score and provide preliminary outcome estimates for larger trial planning. Figure 6 and Figure 7 display overall study design and process.

### 2.3. Screening and Eligibility

Neonates in the Neonatal Intensive Care Unit (NICU) will be screened for eligibility. All the following inclusion criteria should be met: corrected gestational age (postmenstrual age) > 34 6/7 weeks; postnatal age < 28 days; on respiratory support with invasive mechanical ventilation, non-invasive mechanical ventilation (including continuous positive airway pressure (CPAP), non-invasive positive pressure ventilation (NIPPV/BIPAP), or high flow nasal cannula (flow rates > 2 L per minute) with F_i_O_2_ ≥ 0.3); and echocardiography findings suggestive of PH (or score > 0 for PH—Figure 5b). If an echocardiogram has not been completed but other inclusion criteria are met without exclusion criteria, then following consent, a research-dedicated echocardiogram can be obtained to determine eligibility.

The following are exclusion criteria: <32 weeks at birth; weight < 2000 g at time of enrollment; severe HRF with oxygenation index (OI) > 35 or SpO_2_ < 75% on F_i_O_2_ = 1.0 on mechanical ventilation for >60 min in spite of correction of reversible factors such as pneumothorax; a condition or congenital anomaly known to be lethal or associated with high likelihood of death during infancy (e.g., chromosomal anomalies such as Trisomy 18 or 13); congenital heart disease other than atrial septal defect (ASD), patent foramen ovale (PFO), patent ductus arteriosus (PDA) or ventricular septal defect (VSD—single or multiple, <2 mm). Patients of varying race and ethnicity will be included. This may result in patients with darker skin pigmentation randomized to the lower SpO_2_ target. Given prior data regarding possible underestimation of SpO_2_ by pulse oximetry in darker skin patients, we will also measure melanin-index, a non-invasive measurement of skin pigmentation, in each patient to take this into consideration if differences in arterial blood saturation and pulse oximetry are noted.

Infants with congenital diaphragmatic hernia (CDH), Trisomy 21, or HIE on therapeutic hypothermia may be included.

### 2.4. Criteria for Exit from the Trial Intervention

Exit from the active intervention phase will occur if the infant meets any of the following criteria (whichever is earlier). Once exit criteria are met, the infant will follow unit or discharge protocols for SpO_2_ targets if needed (such as going home on oxygen, etc.).

Weaned to nasal cannula oxygen with F_i_O_2_ = 0.21 and flow < 2 LPM;Significant deterioration with OI > 35 or preparation for ECLS cannulation;Hospital discharge;Parental decision to withdraw from the study;Provider (attending physician) decision to withdraw from the study;Death.

### 2.5. Randomization

Patients will be randomized in a 1:1 ratio to the higher 95–99% range (intervention) or the lower 91–95% range (standard). The random allocation was generated in blocks (number of treatments per block will be kept confidential to avoid prediction of future patients’ allocation). Randomization was not stratified. To conceal allocation, the randomization list was generated by a statistician and uploaded into the REDCap randomization module, which then reveals assignment as each patient is enrolled.

### 2.6. Intervention

Following enrollment and consent, patients will be randomized to one of two SpO_2_ targets. The preductal (right upper extremity) SpO_2_ will be targeted at either 95–99% (intervention study arm) with alarm limits at 93% and 100% or target 91–95% (comparison standard arm) with alarm limits at 89% and 96% (Figure 8). Since a patient’s SpO_2_ monitor often leads to titration of F_i_O_2_ by nurses and respiratory therapists, the assigned SpO_2_ target range will be posted on the monitor and the patient’s respiratory support device (i.e., ventilator) to remind staff of assigned goal. Respiratory support management such as non-invasive or mechanical ventilation titration will be at the discretion of the treating team.

### 2.7. Blinding

Since the intervention will be administered to critically ill neonates and requires staff to adjust therapies (e.g., F_i_O_2_) to achieve the assigned SpO_2_ range, the treating medical team, investigators, or participants (guardians) will not be blinded. Given prior problems with the use of pulse oximeters with altered algorithms [33], we decided to use standard clinical pulse oximeters without masking for the current pilot trial.

### 2.8. Outcomes

Primary outcome: Change in HRF/PH score (Figure 3) from baseline (on day 1/time of recruitment) to day 7. Echocardiograms and respiratory details will be collected daily between days 1–7, and the twice weekly until meets exit criteria to assign these scores. If the patient is having an echocardiogram for their routine care, that echocardiogram will be used and overread by members of the research team. Otherwise, a limited research-specific echocardiogram will be obtained. The echocardiograms will be reviewed/scored by two research personnel independently using the PH portion of the score. The HRF portion of the score will be timed to correlate with the echocardiogram portion given the potential fluctuations in HRF (i.e., FiO_2_ administered) throughout the day.

Other outcome data collected will include duration of mechanical ventilation, iNO duration, ECLS use, death before discharge, length of stay, discharge with home O_2_ and diagnosis of bronchopulmonary dysplasia (BPD). Follow up at 4, 8 and 12 months will be completed as well.

### 2.9. Sample Size

We set a target sample size of 30 patients (15 patients per arm), which will permit us to address each of our aims, even if 20% of infants are lost to a particular analysis due to missingness. In particular, we would be able to estimate inter-rater reliability of the day-7 HRF/PH score with sufficient precision to provide 90% power to detect a true inter-rater correlation ≥ 60% and we would have 84.6% power to detect within-infant, over-time intra-cluster correlation coefficients (ICC) ≥ 30% [34].

### 2.10. Data Monitoring Committee

A data safety monitoring board of neonatologists and a statistician will review de-identified, aggregated data every 6 months or after the first 10 patients complete the study if enrollment is slow. They will review adverse events. Since this is a pilot study, we do not anticipate power to warrant early termination due to undue harm or clear benefit and thus analyses are not planned for that. However, given prior data considering underestimation of SpO_2_ by pulse oximetry in darker skin pigmented patients, the data monitoring committee will conservatively evaluate for high risk of infants with darker skin, particularly in the lower SpO_2_ target range.

### 2.11. Anticipated Challenges and Limitations

Recruitment of 30 infants with the above specified criteria, particularly the minimum F_i_O_2_ requirement, may be challenging. Due to standard SpO_2_ targets, we anticipate the F_i_O_2_ will be weaned as quickly as possible, thus making some infants no longer eligible when we screen. Varying F_i_O_2_ by the treating team prior to enrollment may also limit enrollment. Some potential mitigation measures may include an average F_i_O_2_ over the determined time period before enrollment. Additionally, the achieved SpO_2_ may vary compared to the targeted SpO_2_ range. We will conduct the analysis based on the intended target SpO_2_ group, but we will also extract data for the actual achieved SpO_2_. The OI and OSI ranges in our proposed score may not always result in the same category assignment. Additionally, OSI can be problematic at higher SpO_2_ (>97%) as the oxygen–hemoglobin dissociation curve reaches the flat portion. Therefore, if both OI and OSI are available, we will preferentially use OI. We will not measure markers of oxygen toxicity to enhance consenting rates, but we will consider such measurements in the definitive future trial. We will report the PaO_2_ data when available from arterial blood gases. For decreased response to iNO in the higher SpO_2_ target range, this may be difficult to note considering response to iNO is often classified based on improvement in SpO_2_ and the SpO_2_ may already be high in this group and thus change very little. Additionally, the relationship between PVR and PaO_2_ is not linear. We will attempt to mitigate this issue by applying appropriate transformations (i.e., power/log/inverse) to independent and/or dependent variables.

## 3. Conclusions

Current commonly practiced SpO_2_ targets for neonates with PH may not be ideal and there are both clinical evidence and animal model data that suggest a higher SpO_2_ target may potentially improve HRF/PH outcomes. Higher oxygen saturations can potentially increase oxygen toxicity and limit effectiveness of iNO. There have been no clinical trials assessing SpO_2_ targets for term and late-preterm infants with PH to date for a problem that needs answered.

## 4. Trial Status

Enrollment started October 2021 and enrollment is expected to take 2 years.

## Figures and Tables

**Figure 1 children-09-00396-f001:**
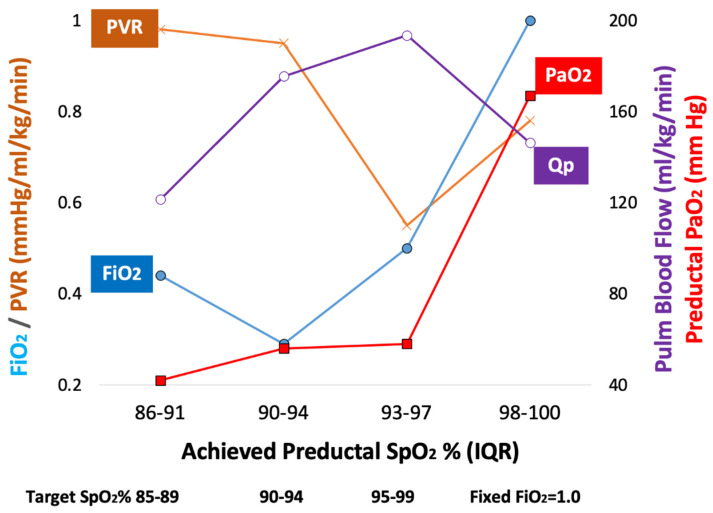
Oxygen saturations targets (SpO_2_), fraction inspired oxygen (F_i_O_2_), pulmonary vascular resistance (PVR), pulmonary blood flow (Qp), preductal partial arterial oxygen tension (PaO_2_) and achieved SpO_2_ (interquartile range—IQR) in lambs with meconium aspiration. The horizontal axis demonstrates the achieved preductal SpO_2_ ranges and interquartile ranges (IQR) closest to the figure and then the target SpO_2_ range just below the achieved ranges. Modified from Lakshminrusimha S., Keszler M., Chapter 34, Diagnosis and management of persistent pulmonary hypertension in assisted ventilation of the neonate, 7th edition, 2021. Copyright Satyan Lakshminrusimha.

**Figure 2 children-09-00396-f002:**
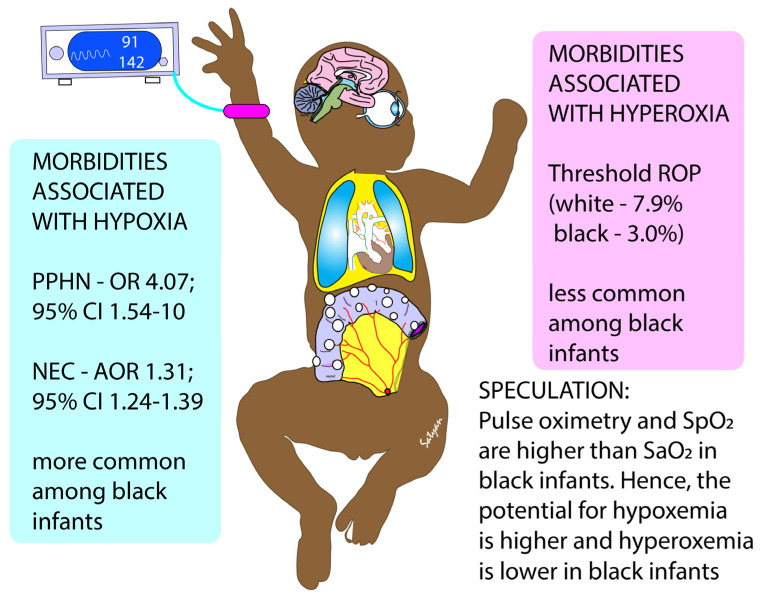
Speculation regarding the basis for increased incidence of morbidities associated with hypoxia among black neonates and those associated with hyperoxia among white infants. Copyright Satyan Lakshminrusimha.

**Figure 3 children-09-00396-f003:**
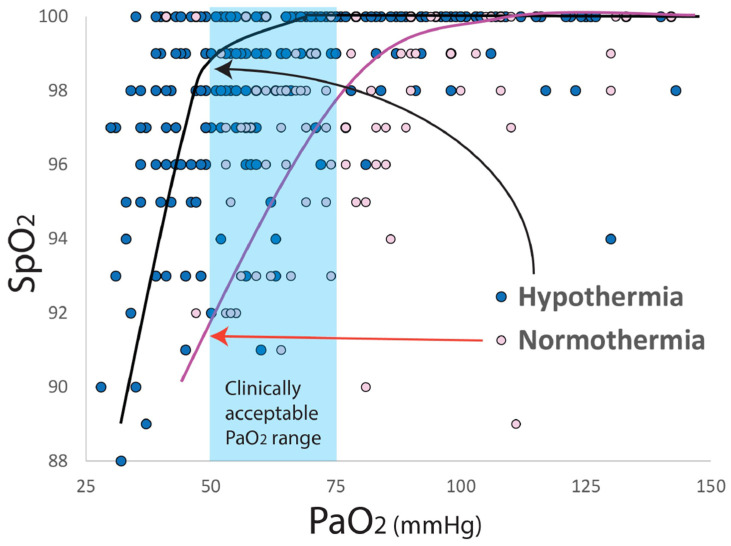
Hemoglobin–oxygen dissociation curve shifts to the left in hypothermia (Afzal et al., 2019). PaO_2_ = partial arterial oxygen tension, SpO_2_ = oxygen saturation. To achieve the clinically accepted range of PaO_2_ 50–80 mmHg, SpO_2_ target is in the low-to-mid 90 s during normothermia but in the high 90 s during whole body hypothermia. Copyright Satyan Lakshminrusimha.

**Figure 4 children-09-00396-f004:**
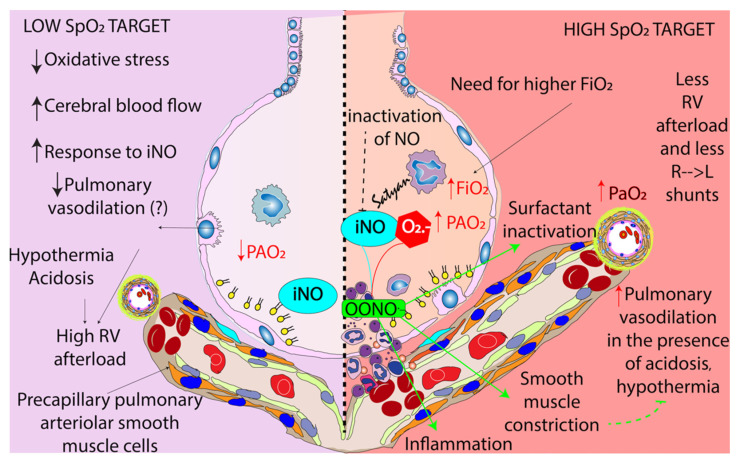
Benefits and risks associated with targeting lower (**left panel**) and higher (**right panel**) oxygen targets. Lower targets may be associated with lower alveolar oxygen tension (PAO_2_), possibly less pulmonary vasodilation especially in the presence of hypothermia and acidosis resulting in exacerbation of hypoxic pulmonary vasoconstriction and high right ventricular (RV) afterload. However, inhaled nitric oxide (iNO) may be more effective due to less inactivation by superoxide anions (O_2_^−^) and there may be less oxidative stress and improved cerebral blood flow. Higher oxygen target may be associated with need for higher FiO_2_, higher PAO_2_ and enhanced pulmonary vasodilation decreasing RV afterload and right-to-left (R → L) shunts. However, higher inspired oxygen can lead to increased superoxide anion formation that may interact with nitric oxide to form peroxynitrite (OONO^−^), a toxic substance that can inactivate surfactant, cause inflammation and pulmonary vasoconstriction. Copyright Satyan Lakshminrusimha.

**Figure 5 children-09-00396-f005:**
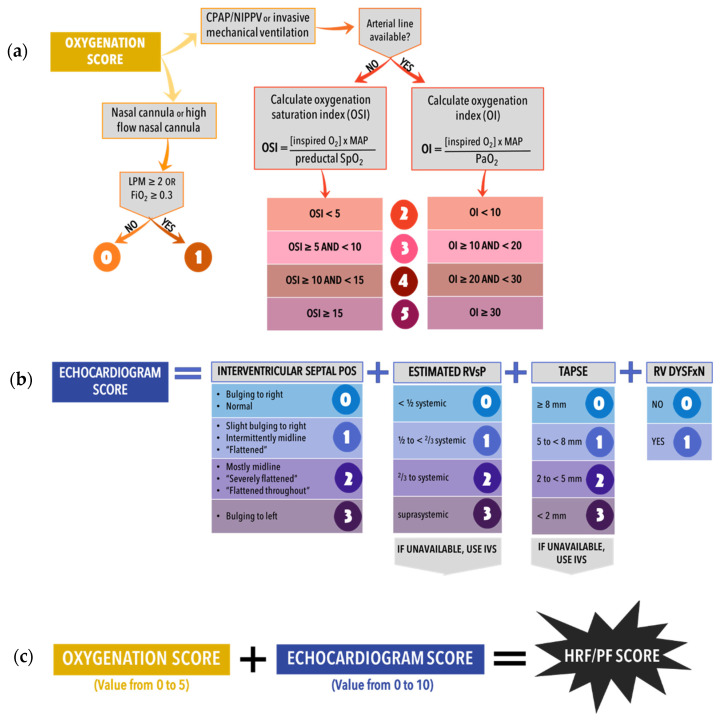
Hypoxic Respiratory Failure and Pulmonary Hypertension Score (HRF/PH). (**a**) Oxygenation component of score. (**b**) Echocardiographic portion of score for PH. (**c**) Total HRF/PH score = Oxygenation score + Echocardiogram Score. CPAP = continuous positive airway pressure, NIPPV = non-invasive positive pressure ventilation, LPM = liters per minute, F_i_O_2_ = fraction of inspired oxygen, OSI = oxygen saturation index, OI = oxygenation index (if both OI and OSI are available, OI will preferentially be used), SpO_2_ = pulse oximetry oxygen saturation, MAP = mean airway pressure, P_a_O_2_ = partial pressure of arterial oxygen, RVsP = right ventricular systolic pressure, RV DYSFxN = right ventricular dysfunction, TAPSE = tricuspid annular plane systolic excursion, mm = millimeters. Copyright Avni Bhatt.

**Figure 6 children-09-00396-f006:**
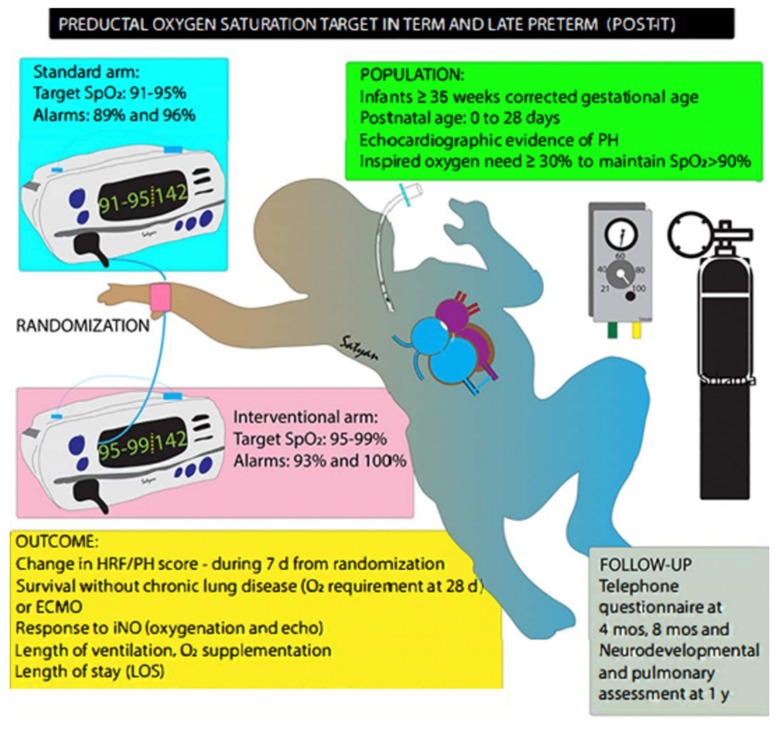
Overview of study design and objectives. Copyright Satyan Lakshminrusimha.

**Figure 7 children-09-00396-f007:**
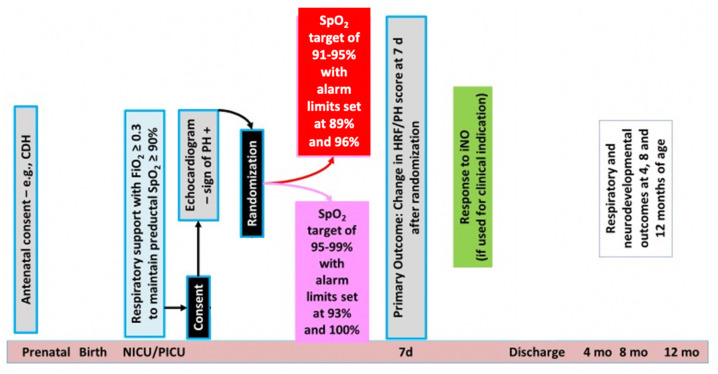
Study events following consent. CDH = congenital diaphragmatic hernia, FiO_2_ = fraction inspired oxygen, SpO_2_ = oxygen saturation, iNO = inhaled nitric oxide. Copyright Satyan Lakshminrusimha.

**Figure 8 children-09-00396-f008:**
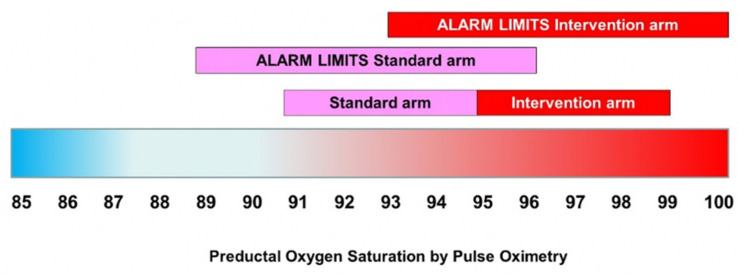
Oxygen saturation target and alarm limits for standard arm and intervention arm. Copyright Satyan Lakshminrusimha.

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
