# Peer review of "Factors to Consider to Study Preductal Oxygen Saturation Targets in Neonatal Pulmonary Hypertension"

_children, 2022, doi:10.3390/children9030396_

Round 1

Reviewer 1 Report

The authors seek to address an important question that continues to hound neonatologists: How much oxygen is the best amount to give to a baby with respiratory disease, in this case, one that includes pulmonary hypertension. They have submitted a manuscript for Stage 1 review that proposes a saturation targeting intervention along with the implementation and evaluation of a novel measurement of pulmonary hypertension, the Hypoxic Respiratory Failure and Pulmonary Hypertension Score (HRF/PH).  The authors provide a detailed, well-referenced background to the pathophysiology (and attendant limitations to our related clinical knowledge) that motivate the proposed study. While over 25% of the references do include the senior author, SL, that is quite appropriate due to his prolific research in this field.

Given the very limited sample size (n = 30 total, 15 in each arm, before dropout), along with the potential inclusion of a wide variety of lung disease, such as CDH*, and possible influences such as hypothermia for HIE, it seems highly unlikely that there will be any statistically significant difference in any major clinical outcomes between the two interventions.  One can only hope that there will be enough of a signal in outcome differences to help design future studies of sat targeting in this population.

*Given relatively high mortality and unique pathophysiology of babies with CDH, I would recommend excluding the few such patients that might be available for enrollment.

It has been a long-standing challenge for non-ECMO centers to decide which of the intermittent arterial blood gases available on NICU patients should be used as criteria for transfer; similarly for ECMO centers, we have to choose which ABGs/OIs to use to commit to ECLS.  This problem is greatly magnified when using the equally dynamic, changing measurement of O2 saturation to score HRF.  As HRF scoring is an integral part of 3 of the 4 study aims, it becomes quite important that this score reflects as well as possible the patient’s status on any given day.  For example, if on Day 2 there is initiation of inhaled nitric oxide, do you report the HRF/PH score before or after INO is started.  Do you try to have 24 hour intervals between assessments, or will the timing be more dependent on availability of echocardiographers?  The more standardized the enrollment and daily measurement assessments, the less likely there would be for the introduction of treater bias (which is made more possible by the lack of blinding of the saturation targeting).

I am not so sure that O2 saturation can be utilized to yield an effective OI as proposed for those babies without arterial access.  As quick demonstration of this, assume a baby is on vent and also assume that O2 saturation follows a relatively linear relationship to pO2 (as per Figure 3, estimating an average of normothermic and hypothermic patients).  Using Figure 3, a pO2 of 30 mmHg roughly equates to 90% sat, and a pO2 of 75 mmHg correlates with O2 sat of 98%.  Comparing OI vs. OSI:

  1. a) Low oxygenation (pO2 = 30, O2 sat = 90), mean airway pressure 10, FiO2 50%

OI = 10 X 100/30 = 33.3 (category score of 5)

OSI = 10 X 100/90 = 11.1 (category score of 4)

  1. b) High oxygenation (pO2 = 75, O2 sat = 98%), mean airway pressure 10 cm H20, FiO2 100%

            OI = 10 x 100/75 = 13.3 (category score of 3)

            OSI = 5 X 100/98 = 5.1 (category score of 2)

Best as I could tell by running several examples, OSI is always less than OI, and the categorical score of OSI is often lower than OI by 1. There are different ways to decrease this bias; one approach would be to simply equate any given O2 sat with an estimated pO2 (using a standardized dissociation curve, which could be corrected for hypothermia).  Admittedly that is not perfect solution, but arguably it would be more accurate than the proposed formulation. 

Is it all possible for authors to analyze clinical data from a cohort of HRF term neonates who met proposed eligibility criteria to see how often ABGs are or are not available?  Identifying such a study population could also be helpful in addressing the OI/OSI issue noted above, i.e. one should be able to look at the O2 saturation (assuming it was known to be pre-ductal) at the time an ABG was obtained and an OI calculated.

I will defer to pediatric cardiology colleagues about the methodology and analysis of echocardiographic data as proposed by this study.  Again, given the amount of serial data that will be collected, the study could yield important predictive measures (albeit the prediction certainty would be limited by the study’s small sample size), e.g. once a baby with HRF is past X days of age with Y HRF/PH score or Z echo findings, the outcome of ECMO or death was near zero.

It is unclear what statistical tools the authors will use to determine the relative validity of HRF/PH, especially compared to other markers of illness, such as OI or echo alone.  The authors could use this opportunity to refine their HRF/PH score to improve it its validity, e.g. adjusting the weighting of the score (currently being 1/3 respiratory status:2/3 echo findings, but not sure why?).

Several of the references (26, 27 and 28) seem to be incomplete.

Figure 5 suggests that a baby can be randomized into the study immediately after consent; seems like flow should be meeting oxygenation requirements and having PH on echo, then getting randomized.

Overall, this manuscript describes an important study designed to give more understanding into the care of critically-ill neonates.  Some further clarification and development of the nitty gritty aspects of the study would be helpful for the readers.  This study, once completed and its results analyzed and reported, may provide important clinical insights to inform the design and implementation of a much larger, multi-center study of oxygen saturation targeting.

Reviewer 2 Report

The review by Siefkes et al. is timely and well written. There is indeed a need for RCT to determine the optimum O2sat to be targeted in term/late preterm neonates with HRF and PPHN physiology. The pros and cons are nicely described. The manuscript is a wonderful review of relevant physiology with superb illustrations. I have the following comments to clarify the points further:

  1. Abstract: Succinct; however, presents both benefits of using higher saturation and downside. Since RCT is designed to test efficacy while monitoring the safety, it can be worded better. The way it is written, higher Sat is hypothesized to both lower the need for iNO and also, conversely reduce the responsiveness to iNO. This will be confusing to reader and the second point can be discussed later in the review where the context for potential decrease in response can be given
  2. Fig 1: Black vs. gray text is not clear. Gray numbers look faded on X axis. Recommend using another color
  3. The compelling arguments for keeping higher pulse oximetry measured O2 saturations in black infants (pages 4-5) will make it less likely that black infants will be enrolled in a RCT. Should they be ethically enrolled into such a trial at a target of 91-95% pulse ox sat? This question will be especially raised for infants on hypothermia as discussed on page 5. Should arterial blood measured saturations be required to correlate at least a few values with pulse ox readings before enrolling such an infant in the comparison trial?
  4. The figures are as insightful as the learning points in the text as above. I find them visually appealing and very easy to follow
  5. The low sat target in Fig 4 only shows adverse effects and no positive benefits, which may convey a bias toward success of higher O2 sat. One would expect less ROS, ONOO-, higher CBF? At least one or two of the favorable effects can be mentioned
  6. Page 7, lines 207-208- in the higher saturation group, the apparent decrease in the response to iNO could be that the saturation is already high and increase will be harder to detect. Since the oxygenation response is not linear with the decrease in PAP- when PAP declines slightly below systemic pressure, there will be a large increase in PaO2 with cessation of shunts and additional decreases in PAP would not linearly increase the PaO2. This needs to be mentioned so we don’t conclude wrongly that the oxygenation response to same dose of iNO is smaller in the higher sat group. This is one of the limitations of using PaO2 as a response variable to pulmonary vasodilators in the setting of PPHN physiology
  7. Page 10, lines 321-325 about outcomes- are echocardiograms in study patients done on day 1 and day 7 or daily? Since many PPHN infants on improving trajectory may not get a repeat echo in first week, does the study cover the 2nd echo or is this routinely done in study sites?

Overall, this is a wonderful review and will be educational for the readers.

Reviewer 3 Report

This paper was very interesting to read, and has the potential for great contribution to our field.  The saturation goals in pulmonary hypertension do remain controversial, and your study design was thoughtful, and the development of a scoring system is novel.  I look forward to reading the outcome of your proposed study.

Author Response

Thank you to Reviewer 3 for your thoughtful review. 

Reviewer 4 Report

The proposed study is long overdue, and the proposed design is well-considered and should yield very important information.  There are a few minor changes that should be considered by the authors. 

  1. The inclusion of patients being treated with hypothermia for HIE is problematic. The authors point out that the effect of hypothermia on increased hemoglobin oxygen affinity can lead to oxygen higher oxygen saturation measurements for a given PaO2, then fret over the hypoxemia that results.  Surely the oxygen demand is lower by design, so the presumably lower oxygen delivery (lower cardiac output is also inevitable) may not matter, provided it is sufficient.  Some kind of tissue oximetry, using NIRS to measure oxidized/reduced cytochrome as well as hemoglobin saturation along with lactate measurements would help to untangle this.  This category of patients seems like it requires its own study given the interactions of temperature on PVR, hypoxic pulmonary vasoconstriction, cardiac output, oxygen demand, and oxygen delivery.
  2. The bases for the “expert opinions” from panels in Europe, Canada, and the USA that recommend the 90-95% targe SpO2 range deserve a little more discussion. After all, the authors rightly point out the wide gap between these recommendations and clinical practitioner opinion. If none of these recommendations are based on sound experimental or clinical data, then how were they developed?  This seems to be important to the rationale for the proposed study.
  3. While the disparity between measurements of SpO2 between light- and dark-skinned patients is important to consider, the examples in oxygen-delivery dependent outcomes were confined to preterm infants, who are not the subject of the proposed study. This section could be shortened, and the figure omitted.
